# Fractionation of Enriched Phosphopeptides Using pH/Acetonitrile-Gradient-Reversed-Phase Microcolumn Separation in Combination with LC–MS/MS Analysis

**DOI:** 10.3390/ijms21113971

**Published:** 2020-06-01

**Authors:** Martin Ondrej, Pavel Rehulka, Helena Rehulkova, Rudolf Kupcik, Ales Tichy

**Affiliations:** 1Department of Radiobiology, Faculty of Military Health Sciences, University of Defense in Brno, 500 01 Hradec Kralove, Czech Republic; martin.ondrej@unob.cz (M.O.); helena.rehulkova@unob.cz (H.R.); 2Department of Molecular Biology and Pathology, Faculty of Military Health Sciences, University of Defense in Brno, 500 01 Hradec Kralove, Czech Republic; pavel.rehulka@unob.cz; 3Department of Biological and Biochemical Sciences, Faculty of Chemical Technology, University of Pardubice, 532 10 Pardubice, Czech Republic; rudolf.kupcik@upce.cz

**Keywords:** phosphopeptides, fractionation, acetonitrile, gradient, enrichment, mass spectrometry, titanium dioxide

## Abstract

Mass spectrometry (MS) is a powerful and sensitive method often used for the identification of phosphoproteins. However, in phosphoproteomics, there is an identified need to compensate for the low abundance, insufficient ionization, and suppression effects of non-phosphorylated peptides. These may hamper the subsequent liquid chromatography–mass spectrometry/mass spectrometry (LC–MS/MS) analysis, resulting in incomplete phosphoproteome characterization, even when using high-resolution instruments. To overcome these drawbacks, we present here an effective microgradient chromatographic technique that yields specific fractions of enriched phosphopeptides compatible with LC–MS/MS analysis. The purpose of our study was to increase the number of identified phosphopeptides, and thus, the coverage of the sample phosphoproteome using the reproducible and straightforward fractionation method. This protocol includes a phosphopeptide enrichment step followed by the optimized microgradient fractionation of enriched phosphopeptides and final LC–MS/MS analysis of the obtained fractions. The simple fractionation system consists of a gas-tight microsyringe delivering the optimized gradient mobile phase to reversed-phase microcolumn. Our data indicate that combining the phosphopeptide enrichment with the microgradient separation is a promising technique for in-depth phosphoproteomic analysis due to moderate input material requirements and more than 3-fold enhanced protein identification.

## 1. Introduction

Reversible protein phosphorylation, one of the crucial post-translational modifications, alters the structural conformation of proteins, affecting their role in cell signaling networks. Overall, phosphorylation controls the protein functions and their subcellular localization and targeting, as well as signal transduction and protein degradation [1,2]. Although phosphorylation is the most frequent post-translational modification of proteins, the number of phosphoproteins to the whole proteome is still relatively low [3].

Phosphoproteomics is a branch of proteomics that identifies, characterizes, and quantifies proteins possessing phosphate groups in their structure. It often uses liquid chromatography–mass spectrometry/mass spectrometry (LC–MS/MS) as a sensitive and specific detection method that is robust both for identification and quantification of phosphoproteins, including localization of present phosphorylation in their primary structure [4,5]. The main advantage of the MS-based phosphoproteomics is the ability to analyze a large number of proteins at once. The use of a high-resolution LC–MS/MS method for this purpose has been particularly advantageous in this regard. However, due to the relatively low stoichiometry of phosphoproteins in the whole proteome, specific methods leading to the enrichment for phosphopeptides in the bottom-up phosphoproteomics are necessary for a successful analysis.

Several studies describing various approaches of phosphoproteins or phosphopeptides enrichment have been published so far [6,7,8]. These approaches include immunoprecipitation, immobilized metal affinity chromatography (IMAC), metal oxide affinity chromatography (MOAC) using titanium dioxide (TiO_2_) or zirconium dioxide (ZrO_2_), sequential elution from IMAC (SIMAC), and more [7].

Enrichment for phosphopeptides using MOAC based on TiO_2_ is probably the most widely regarded approach because of the outstanding enrichment performance and the simple manipulation of the TiO_2_ resins [9]. In 2004, Pinkse et al. introduced a promising and now prevalent strategy for the selective phosphopeptide enrichment with the microspherical TiO_2_ particles (Titansphere^®^) prior to LC–MS/MS analysis [10]. They loaded a peptide sample onto a column filled with TiO_2_ microspheres in 0.25 M acetic acid (pH 2.9). After that, they washed away unbound, non-phosphorylated peptides, and subsequently eluted the phosphopeptides from the column using an alkaline buffer (pH 9) [10].

Although other successful phosphopeptide enrichment procedures provide an improved phosphopeptide identification [11], simple detection of phosphopeptides is not sufficient for an in-depth phosphoproteomic analysis. In this regard, appropriate improvements in the relative quantification of phosphopeptides come to the forefront. There are many studies that review possible approaches in the relative quantification of peptides in phospho/proteomic analysis [12,13,14]. The common enhancing strategy of all label-based quantification methods is phospho/peptide fractionation prior to LC–MS/MS analysis. In general, fractionation significantly increases the proteome coverage due to the higher number of peptides identified and quantified by LC–MS/MS. In fact, there are several fractionation methods used in phospho/proteomics [15,16,17]. One of the most employed is high-pH-reversed-phase (high-pH-RP) fractionation. High-pH-RP fractionation is preferred due to its robustness and high resolution of quantified peptides [18]. However, in the conventional setting of the high-pH-RP fractionation, many phosphopeptides do not efficiently bind to the RP under the high pH conditions; they remain in the flow-through fraction, and attention has to be paid not to lose them during the analysis.

In our study, we modified the high-pH-RP fractionation setup for simple microgradient separation apparatus based on C18 microcolumns used in our laboratory. For this purpose, we have adopted the microgradient elution system previously used in capillary electrochromatography [19], RP separation of peptides [20], glycopeptides [21,22,23], permethylated glycans [24] and in on-line connection with nano-electrospray ionization-mass spectrometry (nanoESI-MS) for analysis of peptide mixtures [25]. We optimized this microgradient elution system for the fractionation of phosphopeptides. For the phosphopeptide enrichment, we used the 5 µm TiO_2_ microspheres (Titansphere^®^), although other enrichment strategies for phosphopeptides could also be applied here. Figure 1 depicts a diagram of our workflow. Compared to the conventional high-pH-RP fractionation, where peptides are usually eluted with the acetonitrile (ACN)-gradient mobile phase under high pH, we used a gradient of both pH and ACN in the mobile phase during the phosphopeptide elution from RP microcolumn. Thus, the goal of our study was to optimize the conditions for simple microgradient separation to increase the number of identified phosphopeptides when compared to a purified sample only. Applying our optimized protocol, we obtained significantly enhanced coverage of identified phosphopeptides (approximately 3-fold increase) compared to the non-fractionated sample. Besides, our in-house fractionation technique is fully repeatable, easy to use, and low-cost in comparison with other fractionation methods requiring expensive instrumentation.

## 2. Results

### 2.1. Conventional High-pH-RP Fractionation Was not Suitable for Phosphopeptides

Phosphopeptide fractionation in quantitative phosphoproteomics provides better coverage of identified and quantified phosphopeptides. In this respect, we have developed a simple in-house technique enabling easy fractionation of enriched phosphopeptides. This technique is based on a microcolumn filled with RP C18 microparticles tightly attached to the gas-tight microsyringe. For comprehension and a full description of our in-house made microcolumn apparatus, see the description in Appendix A.

The high-pH-RP fractionation is a standard fractionation method for peptides in proteomics [26,27]. Considering that our original idea was to apply the microcolumn apparatus to fractionate TiO_2_-enriched phosphopeptides according to the high-pH-RP fractionation protocol, similarly to how it is commonly performed for standard complex peptide mixtures, we used the conventional high-pH-RP mobile phase with the gradient concentration of ACN ranging from 2% to 40% to elute the phosphopeptides in 17 fractions. For the complete composition of the conventional high-pH-RP mobile phase, see Table 1.

HPLC analysis with UV detection of all 17 fractions showed that many peptides eluted in the first fraction (flow-through). LC–MS/MS analysis identified 1580 phosphopeptides and 363 non-phosphorylated peptides in the first fraction, whereas only 85 phosphopeptides from this fraction could also be detected in all of the following fractions (Appendix A). In this case, we concluded that using the high-pH-RP fractionation protocol for enriched phosphopeptides is not suitable due to the basic pH character of the mobile phase leading to elution of a significant portion of phosphopeptides within one fraction (Figure 2).

### 2.2. pH/ACN-Gradient-RP Procedure Provided Improved Phosphopeptide Fractionation

Considering that the high pH causes elution of a significant portion of phosphopeptides into one fraction, we made subtle modifications to our former protocol and used the pH/ACN-gradient-RP fractionation. The conventional high-pH-RP fractionation utilizes fixed pH conditions with the gradient of ACN that is usually applicable for peptide fractionation. On the contrary, our pH/ACN-gradient-RP fractionation technique employed a simultaneous gradient of both pH (3.9–6.6) and ACN (4–24%). We specifically optimized it for the fractionation of phosphopeptides obtained from TiO_2_ enrichment. Table 1 compares the composition of the conventional high-pH-RP and our pH/ACN-gradient-RP elution mobile phases.

As previously, at first, we analyzed the purified sample and all 17 fractions by HPLC. The purified phosphopeptide sample provided a typical LC–UV chromatogram (Figure 3A). Chromatograms of the fractions from the fractionated sample indicated successful fractionation of the TiO_2_-enriched phosphopeptides (Figure 3B). We observed that fractions 6 to 16 were evenly distributed and contained approximately the same absorbance peaks referring to a roughly similar amount of peptide material included in each fraction. Fraction 17 yielded a high-intensity chromatogram, indicating the elution of the rest of the peptide material towards the end of the chromatographic fraction (Figure 3B). For the detailed comparison of the absorbance peaks of individual fractions among replicates, see chromatograms in Appendix A.

Regarding the fractionated sample, fraction 17 appeared to contain a considerably higher amount of peptide material than the rest of the fractions, questioning the elution of the last fraction in a larger volume. However, the overall number of phosphopeptides eluted in each replicate followed the normal distribution within each fraction with a decreasing tendency at the end of the fractionation (Figure 3C).

### 2.3. LC–MS/MS Analysis Confirmed a Significant Improvement in Phosphopeptide Fractionation

We performed the LC–MS/MS analysis with the proteomic identification: (i) to compare the exact numbers of phosphopeptides identified after purification and fractionation and (ii) to verify the applicability of phosphopeptide fractionation for phosphoproteomics. We analyzed both samples in three biological replicates.

For the purified sample, we identified 4554, 4130, and 3873 peptides (including non-phosphorylated ones) in replicate one, two, and three, respectively, with the coefficient of variation (CV) of 8.22%. For the fractionated sample, we identified 10,128, 12,647, and 10,653 peptides in replicate one, two, and three, respectively, with the CV of 11.93% (Figure 4A–C). These values correspond to 2.22-fold, 3.06-fold, and 2.75-fold increase in peptide identification in the first, second, and third replicate, respectively. Regarding the common peptides shared in both purified and fractionated samples, we identified 3405, 3375, and 3069 peptides in replicate one, two, and three, respectively (Figure 4A–C).

In parallel, for the purified sample, we identified 2286, 1904, and 1764 phosphopeptides in replicate one, two, and three, respectively, with the CV of 13.61%. For the fractionated sample, we identified 5295, 6673, and 6177 phosphopeptides in replicate one, two, and three, respectively, with the CV of 11.54%. These values correspond to 2.32-fold, 3.50-fold, and 3.50-fold increase in phosphopeptide identification in the first, second, and third replicate, respectively. Regarding the common phosphopeptides shared in both purified and fractionated samples, we identified 1693, 1645, and 1484 peptides in replicate one, two, and three, respectively (Figure 4A–C). In summary, LC–MS/MS analysis confirmed the significant increase in the peptide (more than 2.5-fold) and phosphopeptide identification (more than 3-fold) in the fractionated sample compared to the purified one (Figure 4D). For summary statistics, see Appendix A.

## 3. Discussion

Phosphoprotein enrichment methods are essential for state-of-the-art phosphoproteomic analysis. MOAC is a powerful enrichment method allowing selective ion exchange between the phosphate group and a metal oxide matrix [28]. Although other enrichment approaches in an optimized setup were successfully introduced for phosphoproteomic research [11], TiO_2_-based MOAC is probably the most widespread enrichment method. This is mainly due to the robustness of the protocol, selectivity of TiO_2_ for phosphopeptides, and also due to the commercial availability of various TiO_2_ materials such as TiO_2_ microparticles (Titansphere, GL Sciences, and Sachtopore-NP, Sachtleben Chemie GmbH), pre-packed TiO_2_ tips (Pierce™ Spin Tips, Thermo Scientific and Glygen NuTips, Glygen Corp.), TiO_2_ columns for on-line applications (TiO_2_ Nano-Trap Columns, Thermo Scientific), and more.

For more in-depth analysis in phospho/proteomics in general, it is inevitable and also a good practice to fractionate the complex peptide mixtures as it improves peptide identification and also quantification [15,29]. However, in the case of phosphoproteomics, there is a dilemma of whether it is better to fractionate first and consequently enrich each fraction for phosphopeptides or to enrich the whole sample for phosphopeptides and fractionate the enriched sample afterward. In general, both approaches are applicable. In the first case, a more selective enrichment may be achieved, while the latter one may be less elaborate, yet still adequately efficient.

Strong cation exchange (SCX) is a traditional first-dimensional separation method. It was widely used as a pre-enrichment fractionation method, although it often suffers from the low resolution of separations [30]. In this regard, hydrophilic interaction chromatography (HILIC) and electrostatic repulsion hydrophilic interaction chromatography (ERLIC) have been developed. Nowadays, they are both commonly applied for pre-fractionation prior to phosphopeptide enrichment [31,32]. Hydrophilic strong anion exchange (hSAX) has also been used successfully for the fractionation of peptide mixtures and phosphopeptides [11,33]. High-pH-RP fractionation is an alternative method providing several practical advantages—robustness, proper separation resolution, and compatibility with further ESI–LC–MS analysis. It typically operates with mobile phases buffered around pH 10 with the gradient of an organic solvent such as ACN. Currently, offline high-pH-RP fractionation has been routinely applied prior to phosphopeptide enrichment by IMAC [34,35] or MOAC using the TiO_2_ approach [27].

In our study, we developed a protocol for a simple microgradient separation of phosphopeptides based on C18 microcolumns. This protocol followed the principles of the conventional high-pH-RP peptide fractionation, but it was significantly modified for efficient fractionation of enriched phosphopeptides. It currently enables enrichment for phosphopeptides before their fractionation, which substantially simplifies the entire workflow and extends the degree of phosphoproteome analysis straightforwardly.

The starting setup of phosphopeptide fractionation with our microgradient separation apparatus was based on the commonly used proteomic protocol for high-pH-RP fractionation of complex peptide mixtures [18] and phosphoproteomic analyses [27,36]. We loaded the sample dissolved in a mobile phase buffered to pH 10 to the microcolumn. The flow-through then represented the first fraction, and further fractions eluted from the gradient. Although the fractionated phosphopeptides were successfully detected also in fractions 2 to 10, many of them eluted in the first fraction because a substantial part of phosphopeptides does not retain effectively on the RP under basic pH conditions (Figure 2). The fraction 1 (flow-through) contained about 25% of all identified phosphopeptides (Appendix A), and thus, these phosphopeptides have to be carefully processed as well to avoid undesirable losses.

This finding is in agreement with our previous experiences with SPE phosphopeptide purification, where we observed a similar lack of retention for some phosphopeptides in solutions with higher pH and generally expected lower retention of acidic compounds under basic pH conditions [37]. Moreover, it is obvious that the application of ACN at a concentration of up to 40% in the mobile phase is not necessary for the fractionation of phosphopeptides since we observed the last LC–UV signal already in fraction 10. These experiments led us to a series of modifications and optimizations of the conventional fractionation protocol that involved (i) changes in solvent composition for phosphopeptide sample loading, (ii) introduction of pH gradient into phosphopeptide elution, (iii) narrowing the range of ACN concentration in the elution gradient, and (iv) setting up the suitable elution volumes for particular fractions.

The initial high-pH-RP fractionation involved loading of the sample to a stationary RP in a buffer of pH 10. However, only partially effective capture of phosphopeptides on RP caused by high pH led us to the loading of the phosphopeptide sample immediately after its elution from TiO_2_ acidified to pH 2. In this case, the phosphopeptide sample was efficiently loaded onto the microcolumn with the simultaneous simplification of the sample preparation step. Moreover, this modification also minimizes the hydrolysis of phosphopeptides under alkaline conditions [38,39]. To optimize the phosphopeptides elution, we adjusted the pH of the mobile phase to a gradient with pH ranging from 3.9 to 6.6. In parallel with the pH adjustment, the ACN gradient increased from 4% to 24%. Such optimization led to more evenly distributed phosphopeptide fractions. Although the last fraction collected a greater part of the elution volume (6 µL of 24 µL) and the LC–UV analysis showed a significant portion of peptides, the number of identified phosphopeptides was similar to the previous fractions (Figure 3C).

Although the microgradient elution system could be automated for obtaining an excellent chromatographic reproducibility [40], the straightforward sample processing and the simplicity of the introduction of the presented manual microgradient elution system in the laboratory are important advantages of such an approach. The direct comparison of corresponding fractions for all three sample preparations in this work is shown in the Appendix A, where the UV chromatograms for each fraction are presented. It is evident that the phosphopeptides are not significantly shifted in their elution across the fractions for different sample fractionations. For this reason, we assume that our protocol is sufficiently reproducible for phosphopeptide fractionation in phosphoproteome studies.

Some may argue that compared to the purified sample, the LC–MS/MS analysis run time of the fractions rises inadequately while the fold change increase of the identified phosphopeptides does not correspond to the prolonged instrument analysis time. In that case, it may be worth considering applying a suitable pooling scheme for particular fractions to decrease the time for the LC–MS/MS analysis. However, a certain decrease in the number of identified phosphopeptides may be expected then, and a balanced trade-off for a particular study has to be selected. Hence, it is therefore up to the eventual end-user to decide whether to pool individual fractions to shorten the analysis time or not.

Taken together, using our in-house microgradient separation apparatus with the properly optimized elution schema, we obtained an optimal distribution of TiO_2_-enriched phosphopeptides over several fractions with a more profound phosphoproteome coverage.

## 4. Materials and Methods

### 4.1. Cell Cultures and Cultivation

We obtained non-small cell lung carcinoma cells (H1299) from the American Type Culture Collections (Manassas, VA, USA). We cultured the cells at 37 °C in a humidified incubator under controlled 5% CO_2_ atmosphere and maintained them in RPMI medium 1640 (Gibco, Paisley, UK) supplemented with a 10% fetal bovine serum, 150 Ul·mL^−1^ penicillin and 50 mg∙mL^−1^ streptomycin (all from Sigma-Aldrich, St. Louis, MO, USA). The maintenance culture was divided twice per week by dilution to a concentration of 2 × 10^5^ cells/10 mL.

### 4.2. Cell Treatment and Recovery of Proteins

We irradiated the cells, placed in a flask using a ^60^Co gamma-ray source with a dose rate of 1.04 Gy∙min^−1^, to stimulate phosphorylation within the cell culture. After irradiation, we transferred the cells back into an incubator and cultivated further for one hour.

After irradiation, we lysed H1299 cells using lysis buffer with a composition of 50 mM ammonium bicarbonate, 1% sodium deoxycholate, and phosphatase inhibitor cocktails 2 and 3 (Sigma-Aldrich, St. Louis, MO, USA) by heating them to 100 °C for 5 min followed by cooling on ice and addition of 1.5 mM MgCl_2_ and 2.5 U∙mL^−1^ Benzonase^®^ Nuclease (Sigma-Aldrich, St. Louis, MO, USA) to degrade all forms of DNA and RNA. We precipitated the obtained proteins using the chloroform/methanol precipitation [41], desalted them using C18 solid-phase extraction cartridges (3M™ Empore™, St. Paul, MN, USA), reduced by dithiothreitol, alkylated by iodoacetamide (both Sigma-Aldrich, St. Louis, MO, USA), and digested with trypsin (Sequencing Grade Modified Trypsin, Promega Corporation, Madison, WI, USA) overnight at 37 °C.

### 4.3. Enrichment for Phosphopeptides

After digestion, we extracted redundant sodium deoxycholate by ethyl acetate [42] and enriched phosphopeptides using titanium dioxide chromatography [43]. We performed the enrichment for phosphopeptides using Mobicol spin columns (MoBiTec GmbH, Goettingen, Germany).

At first, we supplemented samples with trifluoroacetic acid (TFA; 2% *v*/*v*) and glutamic acid (100 mM). Then, we added titanium dioxide microspheres (Titansphere^®^ 5 μm particles, GL Sciences, Torrance, CA, USA) suspended in loading buffer (LB; 65% ACN, 2% TFA, 100 mM glutamic acid) to the samples (peptide:TiO_2_ ratio, 1:10) and incubated under continuous mixing for 30 min at room temperature. Afterward, we washed the microspheres with 200 μL of LB, 200 μL of washing buffer 1 (WB1; 65% ACN with 0.5% TFA), 200 μL of washing buffer 2 (WB2; 65% ACN with 0.1% TFA), and again with 100 μL of WB2. Then, we eluted phosphopeptides by 150 μL of elution buffer (EB; NH_4_OH, pH 11.5).

In order to fractionate enriched phosphopeptides using the conventional high-pH-RP method, we desiccated the eluted sample in an Eppendorf™ Concentrator Plus (Eppendorf, Hamburg, Germany) and dissolved in 2% ACN/20 mM ammonium formate (pH 10). Subsequently, we loaded such a prepared sample directly to the microcolumn.

In the case of the samples enriched for phosphopeptides that were going to be purified and fractionated using our pH/ACN-gradient-RP fractionation, respectively, we acidified the phosphopeptides with 10% TFA to pH 2, divided them into two groups, and continued with purification and fractionation, respectively.

### 4.4. Purification of Phosphopeptides

In the beginning, we prepared our capillary microcolumn and the purification/fractionation apparatus. The microcolumn was composed of the fluorinated ethylene propylene tubing (1/16” × 0.25 mm ID; Valco Instruments Co. Inc. and VICI AG+) and stationary phase particles (Kinetex EVO C18 2.6 µm core-shell particles, Phenomenex Inc., Torrance, CA, USA) suspended in ACN. The purification/fractionation apparatus was composed of a laboratory stand and the 25 µL SGE Gas Tight Syringe (Trajan Scientific and Medical, Melbourne, Victoria, Australia). For a more in-depth description of our microcolumn apparatus, see Appendix A.

The purification itself consisted of 5 steps: 1. wetting, 2. equilibrating, 3. sample loading, 4. washing, and 5. elution. We performed wetting by 25 µL of 80% ACN/0.1% TFA and equilibrating by 25 µL of 2% ACN/0.1% TFA. We loaded the acidified sample onto the column and discarded the flow-through solution. Then, we washed the loaded sample by 25 µL of 2% ACN/0.1% TFA and eluted the purified phosphopeptides by 25 µL of 40% ACN/0.1% TFA. The purified phosphopeptides were immediately desiccated in an Eppendorf™ Concentrator Plus (Eppendorf, Hamburg, Germany).

### 4.5. Fractionation of Phosphopeptides

Similarly to phosphopeptide purification, we also performed fractionation in 5 steps: 1. wetting, 2. equilibrating, 3. sample loading, 4. washing, and 5. elution. The first two steps were consistent with the purification.

In the case of the conventional high-pH-RP strategy, we loaded the sample dissolved in 2% ACN/20 mM ammonium formate (pH 10) directly to the microcolumn and collected the flow-through fraction. Afterward, we washed the sample by 25 µL of 2% ACN/20 mM ammonium formate (pH 10). Before elution, we prepared six solutions of the conventional mobile phase according to Table 1 (conventional elution mobile phases). We aspirated four microliters of each solution into the microsyringe in order from one to six. We then collected the eluates in tubes in the following volumes: 1st fraction—flow-through, 2nd–17th fraction—1.5 µL. The obtained fractions of eluted phosphopeptides were immediately desiccated in an Eppendorf™ Concentrator Plus (Eppendorf, Hamburg, Germany).

In the case of the pH/ACN-gradient-RP strategy, we prepared six solutions of the optimized mobile phase according to Table 1 (optimized elution mobile phases). We aspirated four microliters of each solution into the microsyringe in order from one to six. We then collected the eluates in tubes in the following volumes: 1st fraction—3 µL, 2nd–16th fraction—1 µL, and 17th fraction—6 µL. The obtained fractions of eluted phosphopeptides were immediately desiccated in an Eppendorf™ Concentrator Plus (Eppendorf, Hamburg, Germany).

### 4.6. Liquid Chromatography and Mass Spectrometry Analysis

At first, we dissolved both C18-purified and fractionated peptide samples in 20 µL of 2% ACN/0.1% TFA. In the case of LC–UV analysis, we analyzed 1 µL of each sample using the UltiMate 3000 HPLC system (Dionex, USA). This system coupled with UV detection included µ-Precolumn (300 µm × 5 mm, C18PepMap 5 µm 100 Å particles; Dionex, USA) connected to the analytical NanoEase column (100 µm × 150 mm, Atlantis C18 3 µm 100 Å particles; Waters, USA). We separated the peptides using the bilinear gradient of 5–45% ACN/0.1% TFA over 81 min under a flow rate of 360 nL∙min^−1^ and UV detection set to 215 nm. In the end, we collected data and visualized the respective chromatograms using the Chromeleon software (v. 6.80, Dionex, Sunnyvale, CA, USA).

In the case of LC–MS/MS, we separated peptides using the UltiMate 3000 RSLC-nano HPLC system (Dionex, USA) with a trap column (75 µm × 20 mm) packed with 3 µm Acclaim PepMap100 C18 particles and a separation column (75 µm × 150 mm) packed with 2 µm Acclaim PepMap RSLC C18 particles. We performed the separation with the dual linear gradient using 3–44% ACN in 0.1% formic acid over 89 min for the non-fractionated sample and over 63 min for the obtained fractions under the flow rate of 300 nL∙min^−1^. We monitored the separation using the UV detection system at 214 nm. The separation was directly coupled to MS analysis with the QExactive system (Thermo Fisher Scientific, Waltham, MA, USA) in the positive mode with full MS scan (350–1750 *m*/*z*) at 70,000 FWHM, maximum filling time 100 ms, and AGC target 1E6. We selected the top 12 or top 10 precursors in MS/MS at 17,500 FWHM for purified and fractionated samples, respectively, with maximum filling time 100 ms and AGC target 1E5.

### 4.7. Phosphoproteomic Data Processing

We used Proteome Discoverer software (Thermo Fisher Scientific, Waltham, MA, USA, v. 2.4.0.305) to identify the MS/MS spectra. The raw files from the QExactive mass spectrometer were processed within the processing workflow containing spectrum selector, non-fragment filter, top N peaks filter, precursor detector, SequestHT search engine, target decoy PSM validator, and IMP-ptmRS nodes. The parameters for SequestHT database searching were: protein database—UniProt human reference proteome UP000005640 (19 December 2019); enzyme—trypsin; maximum missed cleavage sites—2; min. peptide length—7; precursor mass tolerance—20 ppm; fragment mass tolerance—0.02 Da; weight of b- and y-ions—1; static modifications—Carbamidomethyl/+57.021 (C); dynamic modifications—Oxidation/+15.995 Da (M), Phospho/+79.966 (S,T,Y); dynamic modifications (protein terminus)—Acetyl/+42.011 Da (N-terminus), Met-loss/−131.040 Da (M), Met-loss+Acetyl/−89.030 Da (M); dynamic modifications (peptide terminus)—Gln→pyro-Glu/-17.027 Da (Q). We further processed the search results obtained in the msf file via the consensus workflow containing PSM Grouper, Peptide Validator, Protein and Peptide Filter, Protein Scorer, Protein FDR Validator, Protein Grouping, and Protein in Peptide Annotation nodes. We set the minimum number of peptide sequences with high confidence (strict target FDR 0.01) to 1 because the enriched samples may often contain only a single peptide per protein. For Venn diagrams reports, we used the Proteome Discoverer graphic tools. We deposited the MS proteomic data to the ProteomeXchange Consortium via the PRIDE partner repository with the dataset identifier PXD018663.

### 4.8. Statistical Analysis

We performed a statistical comparison between the number of only purified and fractionated phosphopeptides using an independent two-sample t-test with unequal variances and critical *p*-values equal to 0.005. Microsoft Excel was used for statistical calculations as fold changes and coefficients of variations. Boxplot diagram was prepared using R version 3.6.1 [44].

## 5. Conclusions

In summary, our optimized procedure resulted in a dramatic increase in phosphopeptide identification. We identified a more than 3-times-higher number of phosphopeptides in the fractionated sample compared to the solely purified one.

Of course, our study has certain limitations such as the manual handling of the separation, the limited upper pressure applied during chromatography separation (given by the resistance of the microsyringe), and the prolonged LC–MS/MS analysis time.

On the other hand, we proved that our microcolumn fractionation is a fully reproducible method. Moreover, the most beneficial aspect of our in-house microgradient fractionation apparatus is in the fast and easy setup. Above that, the method is truly inexpensive. Using the C18 microgradient column for fractionation also simultaneously purifies the sample prior to LC–MS/MS analysis.

According to these conclusions, we believe that using pH/ACN-gradient-RP microcolumn fractionation of enriched phosphopeptides could be a suitable strategy to significantly increase the number of identified phosphopeptides when combined with the sensitive LC–MS/MS analysis usually relying on high-resolution instruments such as the quadrupole-Orbitrap hybrid we used in our study. This work contributes to the optimization of enrichment protocols required for in-depth phosphoproteomic studies.

## Figures and Tables

**Figure 1 ijms-21-03971-f001:**
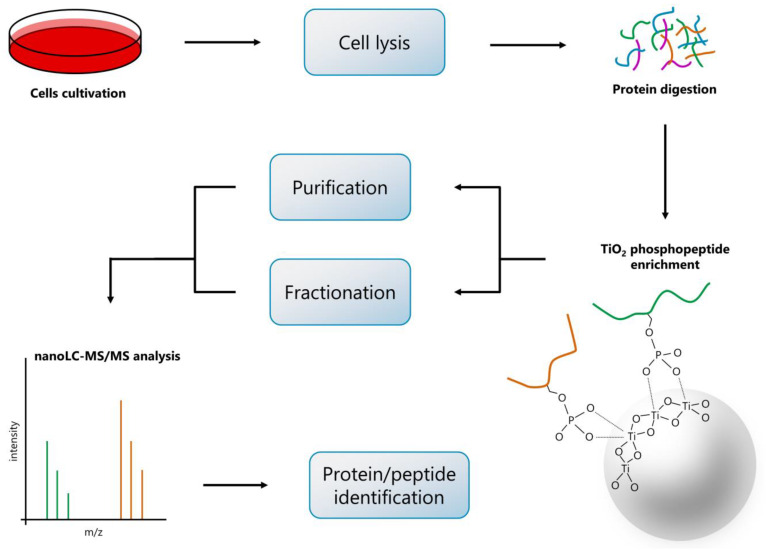
The workflow of sample processing. All samples were processed in the same way except for the step following after the TiO_2_ enrichment, where two types of samples were prepared: 1. TiO_2_-enriched phosphopeptides further purified on C18 reversed-phase (RP) microcolumn (single fraction); 2. TiO_2_-enriched phosphopeptides further fractionated on C18 RP microcolumn using manually formed pH/acetonitrile microgradient (17 fractions).

**Figure 2 ijms-21-03971-f002:**
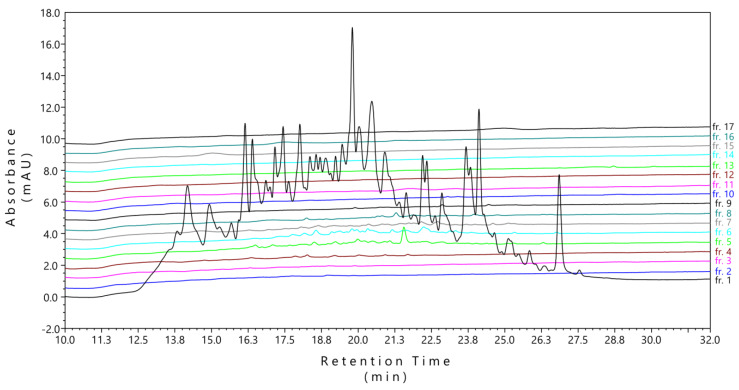
LC–UV chromatograms of the fractionated TiO_2_-enriched phosphopeptide sample obtained from trypsin-digested H1299 cell lysate. The fractionation was performed using the conventional reversed-phase microgradient separation with the acetonitrile (ACN) gradient under basic pH conditions. The first fraction was the flow-through of the TiO_2_-enriched phosphopeptides dissolved in loading buffer of 2% ACN/20 mM ammonium formate (pH 10), and fractions 2–17 were eluted from the microcolumn using the manually formed ACN gradient containing 20 mM ammonium formate buffered at pH 10 (see Table 1 for a detailed composition of the mobile phase used for phosphopeptide fractionation in conventional elution scheme).

**Figure 3 ijms-21-03971-f003:**
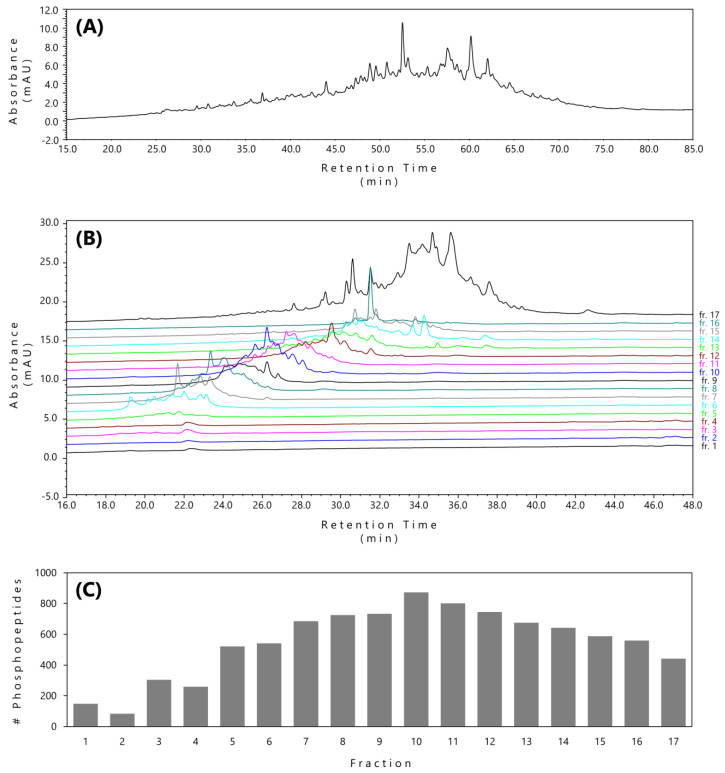
Comparison of the LC–UV chromatograms of purified (**A**) or fractionated (**B**) TiO_2_-enriched phosphopeptide sample obtained from trypsin-digested H1299 cell lysate. The fractionation scheme of the microgradient elution was optimized for a more uniform distribution of peptides throughout the elution fractions (see Table 1 for the detailed composition of the mobile phases used for phosphopeptide fractionation in the optimized elution scheme). Phosphopeptides identified using LC–MS/MS in each fraction were visualized as a histogram (**C**).

**Figure 4 ijms-21-03971-f004:**
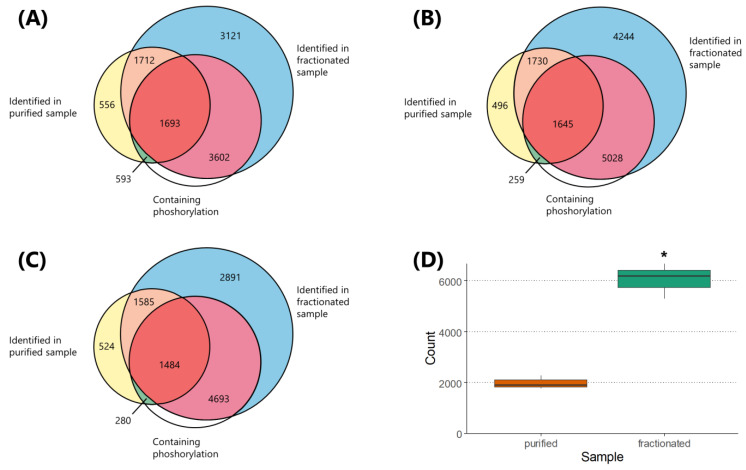
Overview of the peptides identified by the LC–MS/MS analysis. Data from experiments performed within three biological replicates from the H1299 cell line are shown. Venn diagrams of all peptides identified in the first (**A**), second (**B**), and third (**C**) replicate of the purified and fractionated sample with the included number of phosphorylated and non-phosphorylated peptides, respectively. Statistical comparison between phosphopeptides identified in the purified sample and the fractionated sample is visualized as a boxplot of all three replicates (**D**). * Significant difference compared to purified sample (*p*-value ≤ 0.005).

**Table 1 ijms-21-03971-t001:** Comparison of the composition of the conventional high-pH-RP and our optimized pH/acetonitrile (ACN)-gradient-RP elution mobile phases.

		1	2	3	4	5	6
**Conventional elution mobile phase**	ACN	400	320	240	160	80	20
	200 mM HCOONH_4_	100	100	100	100	100	100
	H_2_O	500	580	660	740	820	880
	*ACN concentration*	*40%*	*32%*	*24%*	*16%*	*8%*	*2%*
	*Measured pH*	*10.0*	*10.0*	*10.0*	*10.0*	*10.0*	*10.0*
**Optimized elution mobile phase**	1% acetic acid	100	100	100	100	100	100
	80% ACN/ 200 mM NH_4_OH	300	250	200	150	100	50
	H_2_O	600	650	700	750	800	850
	*ACN concentration*	*24%*	*20%*	*16%*	*12%*	*8%*	*4%*
	*Measured pH*	*6.57*	*5.64*	*5.19*	*4.80*	*4.42*	*3.92*

Conventional high-pH-RP elution mobile phase involved gradient of ACN (2–40%). Optimized pH/ACN-gradient-RP elution mobile phase involved a gradient of both pH (3.92–6.57) and ACN (4–24%). The volume of each component of mobile phases is given in microliters.

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
