# Peer review of "Fractionation of Enriched Phosphopeptides Using pH/Acetonitrile-Gradient-Reversed-Phase Microcolumn Separation in Combination with LC–MS/MS Analysis"

_ijms, 2020, doi:10.3390/ijms21113971_

Round 1
Reviewer 1 Report
In this manuscript the authors developed a protocol for the enrichment of phosphopeptides before (phospho)peptides fractionation. The authors focused on the enrichment by TiO2.
A major complementary method for peptide enrichment is by Fe-IMAC columns as described in this paper https://www.mcponline.org/content/early/2014/11/13/mcp.M114.043109
In order to this paper be accepted for publication it is critical to extend the approach also for the Fe-IMAC approach. In addition, a combined approach, for example, TiO2 followed by Fe-IMAC enrichment (and/or the reverse) and further fractionated.
In section 2.4., the authors eluted the phosphopeptides in ACN/ammonium formate at pH 10. It should be discussed why the solution was not immediately acidified. This procedure is described in https://www.ncbi.nlm.nih.gov/pmc/articles/PMC3002755/
The authors need to provide additional clarification with adopted statistical analysis procedure at section 2.8. Which software has been used ? The detailed approach and some examples should be described in supplementary data.
It is important to include some examples with quantitative information for specific peptide and their phosphopeptide counterpart, including phosphopeptides/peptide ratio before and after the enrichment.
The PRIDE information should include a file describing the meaning and associations of the raw data files with their respective experiments.
Author Response
Dear Referee 1,
Thank you for your time and effort you put into the review of our manuscript. For responses to your comments, please see the attachment.
Martin Ondrej,
The Author

Reviewer 2 Report
I want to congratulation very good manuscript. The new technique of fractionation of TiO2 enriched phosphopeptides is very desirable, hence I am glad that the Authors have taken up this research topic.
The strong suit of the study are:
- creation new method,
- developing a simple microgradient separation of phosphopeptides
- conducting a series of experiments with the aim of modifications and optimizations of this method.
Overall merit is very good. However, some points need to be clarified.
1. It is necessary to provide the purpose of the study (it is theoretically obvious, but it is worth specifying the aim of the research)
2. The software used for statistical analysis should be specified
3. I wonder if it's not worth calculating the coefficient of variation to express the analytical precision of the new method?!
4. It may also be worth mentioning what are the limitations of the study
Author Response
Dear Referee 2,
Thank you for your time and effort you put into the review of our manuscript. For responses to your comments, please see the attachment.
Martin Ondrej,
The Author

Reviewer 3 Report
Ondrej et al. report their development of a new fractionation technique for phosphopeptides after TiO2-based enrichment. For this purpose, a packed microcolumn packed with reversed-phase material is used and the peptides are eluted using a pre-formed gradient in a gas-tight syringe that is pushed through the column. Gradient elution uses a simultaneous change of pH and organic solvent (acetonitrile) concentration, which allows - under optimized conditions - for a roughly even distribution of the peptide elution pattern over many fractions. The performance of the method is compared by analyzing the fractionated sample with an unfractionated sample, each in three replicates. In addition, the authors note that a more conventional high-pH RP fractionation failed because of insufficient binding of the peptides to the RP at high pH, resulting in the elution of the majority of peptides in the flow-through.
Protein phosphorylation is a highly important post-translational modification of proteins and numerous methods have been developed for the high-throughput and large-scale analysis of phosphoproteomes by mass spectrometry. As the authors point out in their manuscript, enrichment by IMAC or MOAC are the predominant techniques, although additional fractionation of the peptide pool either before or after enrichment is helpful for enhanced (phospho)proteome coverage. The approach introduced in this work is insofar interesting as it applied a different solvent system than previous methods.
Although I recommend the manuscript for publication in general, several major and minor points should be addressed in a revised version as summarized below.
Major comments:
1. The reason for the complete failure of the high-pH fractionation step is unclear to me. It cannot be due to a general limitation of phosphopeptide binding at high pH. First of all, this should not affect all phosphopeptides because different phosphopeptides will have different hydrophobicity. Moreover, as can be seen from the results shown in later sections of the manuscript, the enrichment efficiency is far from 100% and therefore a considerable number of unphosphorylated peptides will remain that should behave (=bind) according to expectations. The authors should invest more effort into finding the underlying reason of the failed experiment, especially because high-pH fractionation has been used for phosphopeptide fractionation in the past.
2. It is difficult to make adequate comparisons of highly fractionated and unfractionated samples in terms of phosphoproteome coverage. In this work, 89 min of MS run time was used per unfractionated sample, while 17 x 63 min were used for the fractionated samples. Inevitably, more data will be acquired in the latter setup compared to the former, but in the end the increase of identified peptides by a factor of 2.2 to 3.5 is relatively modest compared to the increase in data acquisition time by a factor of approximately 10. The authors should at least mention this in the text. Ideally, additional data with extended run times for unfractionated samples and fewer fractions (for example pooling after fractionation based on prior LC-UV data) could be compared to find the best compromise.
3. The present setup for the fractionation is difficult to automate and it is difficult to assess the reproducibility of the fractionation step alone from the data presented in this manuscript - the number of identified phosphopeptides varies quite a lot between replicates, but it is not possible to decouple the effect of the fractionation and the LC-MS analysis. Could the authors comment on that?
Minor comments:
Page 3, line 106: simulate > stimulate?
Page 4, starting at line 138: More information about the packed microcolumn would be helpful. How long is the packed bed? How do the particles stay in the tubing? How long does it take to pass the entire volume of solvent through the column?
Page 5, line 209: Data deposition in PRIDE cannot be checked unless access details are provided.
Page 11, line 392: low cost-effectiveness? Should it not rather say high cost-effectiveness (low cost, but high efficiency relative to cost)?
Author Response
Dear Referee 3,
Thank you for your time and effort you put into the review of our manuscript. For responses to your comments, please see the attachment.
Martin Ondrej,
The Author

Reviewer 4 Report
This manuscript describes an analytical method for the fractionation of phosphopeptides derived from enrichment with TiO2 by reversed-phase chromatography followed by high-resolution mass spectrometry. The method includes a newly developed step elution process that involves simultaneous changes of pH and ACN content in each step, whereby the individual elution solutions for each step are stacked as plugs within in a capillary syringe. The proof-of-concept was demonstrated for a peptide sample derived from non-small cell lung carcinoma cells.
The topic of the manuscript is highly relevant to field of phosphoproteomics. The work is innovative and is well conducted. The manuscript is clearly structured and the conclusions are supported by the data.
However, before a publication in International Journal of Molecular Sciences can be considered, this manuscript would require revision considering the two comments listed below.
Comment 1: It is unclear from title and abstract whether the method involves a linear or a step elution process. For greater clarity it is suggested to change the title from “Fractionation of TiO2 enriched phosphopeptides using pH-gradient-reversed-phase microcolumn separation in combination with LC-MS/MS analysis” to “Fractionation of TiO2 enriched phosphopeptides using pH/ACN-step gradient-reversed-phase microcolumn separation in combination with LC-MS/MS analysis”. In addition it is suggested to replace the words “optimized gradient mobile phase to” with “optimized step gradient mobile phase to the” in the abstract on Page 1, Line 23.
Comment 2: The manuscript requires some corrections of the scientific terms and English language e.g.:
Page 1, Line 20: Replace “compliant for” with “compatible with”.
Page 1, Line 37: Replace “group” with “groups”.
Page 4, Line 144: Replace “Ringwood, Australia-Victoria” with “Melbourne, Victoria, Australia”, since Ringwood is a suburb.
Page 5, Line 187 and throughout the manuscript: Italicize “m/z”.
References: Please review subscripts, superscripts and capitalization, e.g. Reference 14: Replace “in Situ” with “in situ”.
Reference 41: Please provide author and editor names.
Author Response
Dear Referee 4,
Thank you for your time and effort you put into the review of our manuscript. For responses to your comments, please see the attachment.
Martin Ondrej,
The Author

Round 2
Reviewer 1 Report
The authors have addressed most of the reviewer's concerns.
Author Response
Dear Referee 1,
Thank you for the revision of our manuscript.
Martin Ondrej,
The Author
Reviewer 3 Report
The authors have significantly improved the manuscript in this revised version. Specifically,
(1) the additional information about the design of the apparatus will help others to reproduce the method,
(2) the detailed analysis of the high-pH fractionation strategy provides important context for comparisons of the two methods,
(3) reproducibility of the fractionation procedure was convincingly demonstrated,
(4) additional commentaries were included in the main text as suggested.
I recommend publication of the article in its present form - with the exception that in line 411, "resilience" should be changed to "resistance".
Author Response
Dear Referee 3,
Thank you for the revision of our manuscript. We have changed the requested word according to your suggestion.
Martin Ondrej,
The Author